# Epidermolysis Bullosa—A Different Genetic Approach in Correlation with Genetic Heterogeneity

**DOI:** 10.3390/diagnostics12061325

**Published:** 2022-05-27

**Authors:** Monica-Cristina Pânzaru, Lavinia Caba, Laura Florea, Elena Emanuela Braha, Eusebiu Vlad Gorduza

**Affiliations:** 1Department of Medical Genetics, Faculty of Medicine, “Grigore T. Popa” University of Medicine and Pharmacy, 16 University Street, 700115 Iasi, Romania; monica.panzaru@yahoo.com (M.-C.P.); vgord@mail.com (E.V.G.); 2Department of Nephrology-Internal Medicine, Faculty of Medicine, “Grigore T. Popa” University of Medicine and Pharmacy, 16 University Street, 700115 Iasi, Romania; 3Department of Nephrology-Internal Medicine, “Dr. C. I. Parhon” Clinical Hospital, Carol I Street, No 50, 700503 Iasi, Romania; 4“C. I. Parhon” National Institute of Endocrinology, 410087 Bucharest, Romania; elenabraha@yahoo.com

**Keywords:** epidermolysis bullosa, mutation, heterogeneity

## Abstract

Epidermolysis bullosa is a heterogeneous group of rare genetic disorders characterized by mucocutaneous fragility and blister formation after minor friction or trauma. There are four major epidermolysis bullosa types based on the ultrastructural level of tissue cleavage: simplex, junctional, dystrophic, and Kindler epidermolysis bullosa. They are caused by mutations in genes that encode the proteins that are part of the hemidesmosomes and focal adhesion complex. Some of these disorders can be associated with extracutaneous manifestations, which are sometimes fatal. They are inherited in an autosomal recessive or autosomal dominant manner. This review is focused on the phenomena of heterogeneity (locus, allelic, mutational, and clinical) in epidermolysis bullosa, and on the correlation genotype–phenotype.

## 1. Introduction

Epidermolysis bullosa (EB) is a heterogeneous group of rare genetic disorders characterized by mucocutaneous fragility and blister formation after minimal trauma [1]. EB presents a variable expression with a wide phenotypic spectrum ranging from localized, mild, acral blistering, and normal life expectancy, to generalized, severe blistering and extracutaneous involvement, which could lead to infections, electrolyte imbalances, or respiratory distress, as well as poor prognosis. Nail dystrophy, keratoderma, and atrophic scarring are common features. Major extracutaneous complications may develop in some subtypes of EB: laryngeal or esophageal stenosis, ectropion, corneal opacification, pseudosyndactyly, and microstomia. Pyloric atresia, nephropathy, muscular dystrophy, cardiomyopathy, and interstitial lung disease, are encountered in rare forms of EB. Some forms of EB (e.g., severe and intermediate recessive dystrophic EB) are associated with an increased risk of developing cutaneous squamous cell carcinoma [1,2].

Four major EB types are described based on the level of skin cleavage: EB simplex (EBS)—with changes at intraepidermal (epidermolytic) level, junctional EB (JEB)—with changes at the intra-lamina lucida (lamina lucidolytic) level, dystrophic EB (DEB)—with changes at the sub-lamina densa (dermolytic) level, and Kindler EB—with multiple changes at the cutaneous level [1]. Precise diagnosis requires the correlation of clinical data with immunofluorescence antigen mapping (IAM), transmission electron microscopy (TEM), and mutational analysis [2]. In EBS the structure and function of keratin intermediate filaments are altered and the intracellular components of the hemidesmosomes are mutated or missing [2]. In JEB, the transmembrane and extracellular proteins of the hemidesmosomes and the anchoring filaments are modified [2]. In DEB, the anchoring fibrils can be absent, reduced in number, or abnormal. In KS, there are multiple cleavage planes (intraepidermal, junctional, or dense sub-lamina) determined by abnormal fermitin family homolog 1, which is a component in the focal adhesion complexes [2].

Research in recent decades has deciphered some of the pathogenic changes in EB, mainly through histological and genetic studies. These researches proved an important genetic heterogeneity and changed the classification of disorders. Our paper is trying to present synthetically the gene mutations and their implications on the cellular level in connection with their clinical features.

## 2. Genes and Proteins Involved in Epidermolysis Bullosa

In EB, cell–matrix interactions are mainly altered. Normally cell–matrix interactions are achieved through two elements: hemidesmosomes and focal adhesion.

Hemidesmosomes (HD) are specialized structures that stably anchor the keratinocytes of the epidermis to the basement membranes. This is done by assembly between the intracellular and transmembrane proteins [2]. HD type I (the classic one) is present in the pseudo-stratified epithelium where interactions between intracellular proteins (plectin, dystonin) and transmembrane protein integrin (α6β4, collagen XVII) and CD151 antigen normally occur. HD type II is found in simple epithelial tissue and consists only of α6β4 integrin and plectin [3].

The focal adhesion is allowed by different proteins: integrin α3β1, transmembrane collagen XIII, and fermitin family homolog 1 (FFH1) [2,4].

Table 1 summarizes the main genes involved in the pathogenesis of EB and the encoded proteins. 

### 2.1. Keratins

Keratins are structural proteins. They form obligate heterodimers assembled into intermediate filaments and 3D cytoskeletons. There are two types of keratins: type I and type II. Keratin-5 (type II keratin protein) and keratin 14 (type I) have an α-helical “rod” domain which is flanked by the head and tail domain [7].

### 2.2. Plectin

Plectin is a cytoskeletal linker protein expressed in skin and skeletal muscle. It links intermediate filaments to hemidesmosomes and in this way functions as a mediator of keratinocyte mechanical stability in the skin [8].

### 2.3. BTB Domain Containing Kelch-like Protein

Unlike most EBS-associated proteins, the Kelch-like protein 24 (KLHL) is not a structural protein. KLHL proteins are expressed almost ubiquitously at low levels and are involved in cytoskeletal organization, the regulation of cell morphology, cell migration, protein degradation, and gene expression. They contain a BTB domain that binds to cullin 3, a scaffold protein required for the ubiquitination and proteasomal degradation of substrate proteins [7,9].

### 2.4. Dystonin

Dystonin is one of the largest human proteins and as a member of the plakin family is a structural component of hemidesmosomal inner plaques in basal keratinocytes, a key attachment point for keratin intermediate filaments and the location for other major plaque proteins, such as plectin [10]. The *DST* gene encodes dystonin; alternative splicing produces multiple tissue isoforms expressed in the central nervous system, skin, heart, and skeletal muscle [11,12].

### 2.5. Exophilin

*EXPH5* gene encodes exophilin-5, an effector protein of the Rab27B GTPase. This protein plays a role in intracellular vesicle trafficking and exosome secretion.

### 2.6. Tetraspanins

Tetraspanins form a protein superfamily widely distributed in the epidermis, renal glomeruli, and proximal and distal tubules. One of its members is the CD151 antigen. In the epidermis, the CD151 antigen participates in the formation of hemidesmosomes and forms very stable laminin-binding complexes with alpha-3beta-11 and alpha-6beta-4 integrins. This allows cell adhesion and the intracellular vesicular transport of integrins. In the kidney, CD151 forms complexes with integrins α3β1 and α6β1 and is essential for the proper assembly of the glomerular and tubular basement membranes [13].

### 2.7. Laminin Subunits

Laminin 332 is a heterotrimeric molecule that consists of alpha-3, beta-3, and gamma-2 subunits, and is essential for the formation and function of the basement membrane. Laminin 332 interacts with integrin alpha-6beta-4, alpha-3beta-1, and collagen VII, and plays a pivotal role in epidermal adhesion, cell survival, migration, and regeneration. The *LAMA3*, *LAMB3*, and *LAMC2* genes encode the alpha-3, beta-3, and gamma-2 chains of laminin 332 [14,15].

### 2.8. Collagen XVII

Collagen XVII is expressed in the epithelial hemidesmosomes of the skin, mucous membranes, and eyes [16]. It consists of three identical α1 chains. It has three major domains: a globular intracellular domain, a transmembrane domain, and an extracellular domain. The extracellular domain consists of 15 collagenous domains (cell adhesion domains) and 16 noncollagenous domains (with roles in triple-helix folding) [16]. The collagen XVII binds intracellularly to plectin, dystonin, and β4 integrin, and extracellularly to α6 integrin (that binds to CD151) and laminin 332 [2].

### 2.9. Collagen VII

Type VII collagen is secreted by keratinocytes as procollagen VII. The procollagen VII is formed by three pro-alpha-1 chains that fold into one molecule. The molecule has an N-terminal noncollagenous 1 (NC1) domain, followed by an extended collagenous domain, and ends with the NC2 domain at the C-terminus. Its role is in anchoring the fibrils that form the cutaneous basement membrane zone adhesion complex [7].

### 2.10. Integrins

Integrins are transmembrane protein complexes, consisting of alpha and beta chain subunits. The alpha-6 beta-4 integrin is involved in hemidesmosome formation and stability and interacts with laminin, plectin, and dystonin. The activation of integrins mediates extracellular cell–matrix interactions and cytoskeleton organization [17]. *ITGA6* encodes the alpha-6, whereas *ITGB4* encodes the beta-4 subunit of the alpha-6beta-4 integrin. The *ITGA3* gene encodes the integrin alpha-3 subunit that is connected with a beta-1 subunit to form an integrin involved in interactions with extracellular matrix proteins including laminins. The integrin alpha-3 subunit is expressed in basal keratinocytes, podocytes, tubular epithelial cells, alveolar epithelial cells, and many other tissues [7].

### 2.11. Fermitin Family Homolog 1

The *FFH1* gene is expressed at the dermal–epidermal junction, oral mucosa, and in the gastrointestinal tract [7,18,19]. It is involved in the connection between the actin cytoskeleton and the extracellular matrix by focal adhesion [4]. Also, FFH1 participates in integrins’ activation [2].

The main interactions between proteins are shown in Figure 1.

## 3. Correlations Genotype–Phenotype

### 3.1. EB Simplex (EBS)

EBS is the most common type of EB accounting for ~70% of all EB [7]. EBS has a prevalence of 6/1,000,000 individuals and an incidence of 7.87 per one million live births [20]. EBS is characterized by skin blistering due to intraepidermal cleavage (within the basal layer of keratinocytes) [7]. In general, blistering is caused by trauma, rarely occurs spontaneously, and tends to heal without scarring. EBS has a variable spectrum of severity ranging from mild blistering of the hands and feet to generalized forms with extracutaneous involvement and is sometimes fatal. Onset varies by subtype and occurs, usually, at birth or during infancy, although patients with localized EBS may not develop their first blisters until adolescence or early adulthood [21,22]. Mutations in the *KRT5* and *KRT14* genes occur in 75% of cases with EBS [23]. The most recent EB classification includes 14 EBS subtypes based on the distribution and severity of the blisters, specific cutaneous lesions, mode of inheritance, affected gene/protein, and extracutaneous manifestations [1].

#### 3.1.1. EBS, Localized

The most common and mildest subtype of EBS is localized EBS, previously known as Weber–Cockayne disease, with a reported incidence of 3.67 per one million live births [20], but probably a significant percentage of mild cases remain undiagnosed. Localized EBS is characterized by the formation of blisters usually limited to the palms and soles of the feet. The lesions can also appear in other regions of recurrent trauma, such as the knees and shins of a crawling toddler or flexures during hot weather. The age of onset is variable, and the lesions frequently develop in infancy/early childhood and are rarely present at birth or appear in adolescence/adulthood. Nail involvement is uncommon. Common complications are secondary infections, especially foot blisters. Lesions worsen in the warmer months and some patients develop focal palmoplantar keratoderma during adulthood [22,24]. Intraoral blisters or ulcerations are seen during infancy and usually are asymptomatic [22,25]. The disorder has an autosomal dominant inheritance and is produced by missense mutations in the *KRT14* and *KRT5* genes [26]. The mutations are located outside the highly conserved boundary motifs of the rod domain, usually in the head, tail, or non-helical portions, including the linker area of keratin. They are most frequently found in clusters including in the non-helical L12 linker motif, in the amino-terminal homologous domain (H1) of keratin-5, or in the 2B segment of keratin-14 [27,28]. Hut et al. proposed a genomic mutation detection system for exons 1, 4, and 6 of *KRT14* that encode the 1A, L1-2, and 2B domains containing the mutation hotspots [29]. Jiang et al. suggest that in localized EB sequences, coding for the head and the non-helical linker regions of *KRT5* should have propriety for the mutation screening [30]. However, mutations were also discovered in the conserved 1A and 2B helix hotspots but with conservative amino acid changes [28]. This is consistent with a report by Cho et al. regarding the influence of polarity on the severity of EBS [31].

#### 3.1.2. EBS, Severe, AD, KRT14/5

Severe EBS, with a reported incidence of 1.16 per one million live births, is characterized by generalized and severe blistering and birth-onset. A suggestive feature is the presence of multiple small blisters in a grouped or arcuate configuration, which explains the previous name “EB herpetiformis”. Hemorrhagic blisters are also present. The involvement of the oral mucosa and nail dystrophy are common. Mucosal involvement may interfere with feeding, especially in neonates and infants. Inflammation can occur in hemorrhagic blisters followed by milia and hypo- and hyperpigmentation of the skin. The lesions tend to improve with age or paradoxically, in some cases, during periods of heat or fever. Progressive confluent palmoplantar keratoderma is common, precocious (childhood-onset), and more severe than in other subtypes. This subtype is frequently associated with marked morbidity and in a minority of cases with neonatal/infancy mortality [22,24,32,33]. Dominant-negative mutations in *KRT14* and *KRT5* have been clustered in regions involved in the highly conserved ends of the rod domains or the helix boundary motifs of keratin. Substitutions are frequently reported and involve highly conserved amino acids within the helix initiation or termination motifs blocking the heterodimerization of keratin polypeptides. Common mutations change the glutamic acid from position 477 of keratin-5 (*KRT5* E477) or the arginine from position 125 of keratin-14 (*KRT14* R125). Both mutations cause the extensive formation of cytoplasmic protein aggregates, a hallmark of severe EBS [22]. Monoallelic in-frame deletion, splice-site, or nonsense mutations were also reported, leading to abnormal proteins with dominant-negative effects. Major changes in polarity or acidity are associated with this subtype [34,35,36]. Around 70% of cases with severe EBS are generated by one mutation in the *KRT5* gene (c.1429G > A; *p*.Glu477Lys or E477K) and three other in the KRT14 gene (c.373C > T [*p*.Arg125Cys or R125C]; c.374G > A [*p*.Arg125His or R125H]; c.368A > G [*p*.Asn123Ser or N123S]) [26]. Vahidnezhad et al. reported a case with digenic inheritance, an association of a mutation in the *KRT5* gene and others in the *KRT14* gene [37].

#### 3.1.3. EBS, Intermediate, AD, KRT14/5

This subtype, previously known as Koebner EBS, has an intermediate phenotype, between localized EBS and severe EBS. Blisters appear at birth or in the first few months with generalized distribution, which are milder than those in severe EBS but without a “herpetiform” configuration. The frequency of milia, scarring, nail dystrophy, and oral lesions is intermediate between that of localized EBS and severe EBS. Focal palmoplantar keratoderma can be observed. Lesions worsen in the warmer months. Lesions tend to improve in adolescence when they may become localized to the hands and feet [7,22,24]. Pathogenic variants in the 1A or 2B segments (except the beginning of 1A, 1B, and the end of 1B, which associate severe phenotype) of the rod domain of *KRT5* and *KRT14* are common in intermediate EBS cases. These mutations do not interfere with the elongation process during filament assembly, so filaments essentially appear normal upon ultrastructural examination but are structurally weakened [38,39].

#### 3.1.4. EBS, Intermediate or Severe, AR, KRT14/5

EBS due to *KRT14* or *KRT5* pathogenic variants is frequently inherited in an autosomal dominant mode but autosomal recessive cases were also reported. Most recessive cases are produced by *KRT14* variants and have a phenotype similar to previously described subtypes, but an improvement in blistering with age is not expected. Focal dyskeratotic skin lesions were also reported. Homozygous mutations in *KRT5* lead to severe phenotype, extracutaneous manifestations, and early mortality. Nonsense, missense, splice site, and deletions in *KRT14* have been associated with recessive inheritance. The unaffected parents of each patient were heterozygous for the respective mutations [40]. Rugg et al. consider that these mutations are likely to be associated with a nonsense-mediated messenger RNA decay leading to a functional “knockout” of keratin-14 [41]. Jonkman et al. suggested that increased expression of keratin-5 has a compensatory effect because keratin-14 knockout mice die within the first few weeks after birth [42].

#### 3.1.5. EBS with Mottled Pigmentation

EBS with mottled pigmentation (EBS-MP) with a reported incidence of 0.07 per one million live births, presents generalized blistering from birth but the severity of the lesions is intermediate. The hallmark feature is mottled or reticulate macular pigmentation typically of the neck, upper trunk, and acral skin. Small hyperpigmented macules appear in early childhood, progress over time, and coalesce into a reticulate pattern. Hypopigmented macules are interspersed. The pigmentation does not occur in areas of blistering and often disappears in adult life. Punctate palmar and plantar keratoderma and nail dystrophy may occur. The majority of cases (more than 90%) presented a missense mutation (c.74C > T [*p*.Pro25Leu or P25L]) in the *KRT5* gene [43]. The pigmentary anomalies observed in this EB form could be correlated with the modification of melanosome transport where the non-helical head domain of keratin-5 is involved [44,45]. However, the pathogenic mechanism is incompletely deciphered and some modifiers could interfere with the function of keratin -5. For example, in some cases the pathogenic mutation c.356T > C (*p*.Met119Thr or M119T) in the *KRT14* gene was identified [46].

#### 3.1.6. EBS, Migratory Circinate

EBS, migratory circinate is a rare subtype, previously known as EBS with migratory circinate erythema. It is characterized by generalized blistering from birth with a background of inflammatory migratory circinate erythema that fades and heals with hyperpigmentation (sometimes with a mottled pattern) but without scarring. Nail dystrophy may occur. Some mutations were reported in this form of EB. For example, Gu et al. reported a heterozygous deletion c.1649delG (*p*.Gly550fs) in exon 9 of the *KRT5* gene which leads to a frameshift and delayed termination codon in two unrelated families with EBS, migratory circinate. Lee et al. identified a de novo in-frame 12-bp deletion in exon 7 of the *KRT5* gene, which alters the 2B domain of keratin-5 [47]. Mutations in the keratin-5 tail domain have been related to EBS with unusual features, such as mottled pigmentation and pigmentary disorders, suggesting a possible role of this domain in the regulation of inflammation and pigmentation [48,49].

#### 3.1.7. EBS, AD, with PLEC Mutations

Previously known as Ogna EBS, this subtype presents birth-onset and mild skin blistering, mainly acral and occasionally widespread. The characteristic features are easy skin bruising with the formation of violaceous and hypopigmented macules. Koss-Harnes et al. found the same mutation in exon 31 of the *PLEC* gene (c.6328C > T [*p*.Arg2110Trp or R2110W]) in two unrelated families with Ogna EBS. This mutation changes the plectin polypeptide, which connects the basal keratins to the hemidesmosomal plaque, and generates an aberrant ultrastructure of hemidesmosomes’ attachment plates and a frequent fragmentation of hemidesmosomes [50]. Bolling et al. identified mutations in *PLEC* in 6/16 of individuals with biopsy-proven EBS who lack identifiable pathogenic variants in *KRT5* or *KRT14* genes. They suggest that *PLEC* mutations may be more common than previously realized [51].

#### 3.1.8. EBS, AR, with PLEC Mutations

This rare subtype with recessive inheritance has a more severe phenotype than the dominant form. EBS with AR *PLEC* mutations is characterized by generalized skin blistering that heals with scarring and hyperpigmentation. Nail dystrophy is severe. Mucous membranes and the heart and muscles are spared [7]. Gostynska et al. identified homozygosity for a nonsense mutation c.46C > T [*p*.Arg16X] in the first exon of the gene encoding plectin isoform 1a, in two sisters from a consanguineous family. Plectin has eight tissue-specific isoforms in humans, arising from the alternate splicing of the first exon. The isoform 1a is not expressed in striated or cardiac muscle tissue, so muscular dystrophy or cardiomyopathy are not expected to develop in these cases [52].

#### 3.1.9. EBS, Intermediate with Muscular Dystrophy

EBS, intermediate with muscular dystrophy (EBS-MD) is an autosomal recessive disorder characterized by early generalized blistering and variable (usually during childhood) onset of progressive limb-girdle type, muscular dystrophy. Considerable variability in the severity of the muscle weakness, sometimes not noticeable until the fourth decade of the patient’s life, is reported. Onychodystrophy, focal plantar keratoderma, and mucosal involvement are common. Abnormal dentition (decay teeth), upper respiratory tract stenosis, urethral strictures, dilated cardiomyopathy, ventricular hypertrophy, and alopecia have been reported [53]. The majority of EBS-MD patients present compound heterozygous or homozygous truncation mutations in exon 31 of the *PLEC* gene, which encodes the rod domain of plectin. Natsuga et al. examined plectin expression in the skin of patients with *PLEC* mutations. In EBS-MD, the expression of the N- and C-terminal domains of plectin remained detectable, although the expression of rod domains was absent or markedly reduced. The alternative splicing of exon 31, resulting in a rodless but still partially functional plectin, was suggested to account for the milder phenotype. Few EBS-MD cases have in-frame mutations in the N-terminal domain of plectin, where the actin-binding domain (ABD) and spectrin repeats are conserved. It is possible that the binding deficits with integrin beta-4 and the collagen alpha-1(XVII) chain, which may explain the phenotype [54,55,56,57].

#### 3.1.10. EBS, Severe with Pyloric Atresia

EBS, severe with pyloric atresia (EBS-PA) presents a severe phenotype with widespread generalized blistering or an absence of skin at birth and pyloric atresia. Antenatally, pyloric atresia can manifest with polyhydramnios. Additional features include failure to thrive, aplasia cutis, anemia, sepsis, intraoral blistering, urethral stenosis, and urologic complications. Death usually occurs in infancy [53]. Immunohistochemical studies showed an absent expression of plectin. In contrast to EBS-MD, EBS-PA patients typically have *PLEC* mutations, nonsense or frameshift, outside of exon 31, which leads to loss of both full-length and rodless plectin. Inheritance is autosomal recessive [58].

#### 3.1.11. EBS, Intermediate with Cardiomyopathy

This subtype is characterized by marked erosions in the limbs at birth, healing with dyspigmentation and cribriform atrophic scars, follicular atrophoderma, and late-onset dilated cardiomyopathy. Keratoderma, milia, nail and oral involvement, and progressive diffuse alopecia are reported [7,34,59]. All cases had a heterozygous gain-of-function in *KLHL24* gene start codon mutation, with c.1A-G being the most prevalent. This mutation produces a truncated KLHL24 protein lacking the initial 28 amino acids (KLHL24-ΔN28). The substrate of the KLHL24 protein is keratin-14 and the more stable KLHL24-ΔN28 due to gain-of-function variants inducing the excessive ubiquitination and degradation of keratin-14. Hee et al. consider that *KLHL24* gene mutations disturb the turnover and degradation of intermediate filaments [60]. Schwieger-Briel et al. showed that KHL24 is expressed at similar levels in keratinocytes and cardiomyocytes and may disrupt the degradation of the structural cytoskeletal proteins involved in mechanical resilience [61]. Hedberg-Oldfords et al. reported familial cases with cardiomyopathy due to *KHLH24* gene mutation with polyglucosan accumulation in some cardiomyocytes and with an accumulation of glycogen, desmin, and tubular structures in the cardiomyocytes and in skeletal muscle fibers. They suggest a pivotal role for KLHL24 during cardiogenesis, based on strong *KLHL24* gene expression in early ventricular myocytes and later in the established heart ventricle. As desmin is the cardiac homologue of keratin-14, Vermeer et al. hypothesized that KLHL24-ΔN28 leads to the excessive degradation of desmin, affecting tissue morphology and function. Also, dominant mutations in desmin are associated with a severe form of cardiomyopathy [9,62,63].

#### 3.1.12. EBS, Localized or Intermediate with Dystonin Deficiency

EBS, localized or intermediate with dystonin (BP230) deficiency presents an early-onset with predominantly acral blistering, larger (several centimeters) than in localized EBS. The blisters appear in areas of mechanical trauma but also in non-pressure-prone sites. Blistering could heal without scarring or with post-inflammatory hypo- or hyperpigmentation. Asymptomatic plantar keratoderma was reported [64]. Loss-of-function mutations in the *DST* gene lead to a complete absence of hemidesmosomal plaques, a loss of adhesion, and increased cell spreading and migration. Reduced integrin beta-4 at the cell surface and increased levels of keratin-14 and integrin beta-1 were detected in abnormal cells. Mild phenotype, in contrast to autoimmune bullous pemphigoid who have autoantibodies to dystonin, could be explained by an upregulation of keratin-14 expression. The inheritance is autosomal recessive, but a semi-dominant transmission mode is also plausible because some heterozygous recall some blistering in childhood [34,64,65].

#### 3.1.13. EBS, Localized or Intermediate with Exophilin-5 Deficiency

EBS, localized or intermediate with exophilin-5 deficiency is characterized by localized or generalized intermittent blistering with onset at birth or in early childhood. Skin fragility improves with age, but lesions could heal with hypopigmentation or mottled pigmentation, especially on the trunk and proximal limbs. Skin atrophy and acral blistering with hemorrhagic crusts are cited. Diociaiuti et al. consider that the lack of extracutaneous and adnexal involvement, together with the modest phenotype, differentiates this subtype from the common dominant EBS-MP due to keratin mutations [7,66]. McGrath et al. reported disruption of the keratin filament network, more cortically distributed F-actin, and significantly reduced cell adhesion in keratinocytes from patients with truncating mutations in *EXPH5*. Monteleon et al. demonstrated that exophilin-5 is involved in the delivery of lysosome-related organelles (LROs) to the plasma membrane and is essential for the differentiation of human keratinocytes. LROs are also involved in the packaging and trafficking of melanin, which may explain the pigmentation anomalies. Nonsense and frameshift mutations with autosomal recessive inheritance were reported [1,67,68,69].

#### 3.1.14. EBS, Localized with Nephropathy

EBS, localized with nephropathy presents early-onset blistering, particularly on pretibial areas associated with nephropathy. Early alopecia, poikiloderma, and nail dystrophy may occur. Involvement of the ocular, oral, gastrointestinal (including esophageal webbing), and urogenital mucosal membranes is reported. Nephropathy manifests with proteinuria and progression to end-stage renal disease [7,34,70]. Homozygous frameshift and splice-site mutations in exon 5 of the *CD151* gene leading to truncated proteins without an integrin-binding domain, were reported [13,53,70,71].

### 3.2. Junctional EB

Junctional EB (JEB) is a disease with different prevalence in different geographic areas. The USA National EB Registry reported a prevalence of 0.49 per one million population, whereas the Dystrophic Epidermolysis Bullosa Research Association of America showed a prevalence of 3.59 per million per one million population. In Germany, the prevalence of disease was estimated at 6.7 per one million population [20,72,73]. Early lethality of severe forms could explain the differences. The incidence is higher in the Middle East, due to the high inbreeding coefficient. The inheritance is autosomal recessive and germline mosaicism and uniparental isodisomy were reported [74,75,76]. In JEB skin cleavage occurs within the lamina lucida of the basement membrane zone. The severity of cutaneous and mucosal fragility varies considerably ranging from forms with early lethality to milder phenotypes. A characteristic feature is represented by mature dental enamel anomalies ranging from small pits in the enamel surface to generalized hypoplasia. Impaired adhesion of the odontogenic epithelium from which ameloblasts are derived is involved in abnormal enamel formation [7,22,25]. On a clinical basis, JEB was divided into several categories.

#### 3.2.1. JEB, Severe

In severe JEB, previously known as Herlitz JEB, extensive mucocutaneous blistering with early-onset (at birth or in the neonatal period) may lead to large erosions with extensive loss of proteins, fluids, and iron, which increases susceptibility to infection and electrolyte imbalance. Sometimes, at birth, blisters may be mild and localized to periungual, buttock, or elbow regions [7,22]. The pathognomonic feature is an exuberant granulation tissue located in orofacial (which produces microstomia), periungual, or friction regions, Accumulation of subglottic granulation tissue may lead to a weak, hoarse cry, stridor, and respiratory distress. Alopecia and mature dental enamel defects are common. Involvement of the mucous membranes of the upper respiratory tract, esophagus, bladder, urethra, rectum, and cornea has been reported. Scarring pseudosyndactyly of the hands and feet with severe loss of function has been cited. JEB has the highest risk of infant mortality among the EB subtypes, and the major causes are sepsis, failure to thrive, or tracheolaryngeal obstruction [7,22,24,25,72]. Biallelic mutations in the *LAMA3*, *LAMB3*, and *LAMC2* genes were identified in severe JEB. Varki et al. reported a high proportion of *LAMB3* mutations. The majority lead to premature stop codons, mRNA decay, and synthesis of no protein or to truncated unstable polypeptides. The most frequent mutation in the *LAMB3* gene (45–63%) is c.1903C > T (*p*.Arg635Ter or *p*.R635X) [77]. The distribution of laminin 332 in multiple epithelial basement membranes, including those of the cornea, kidney, lung, thymus, brain, gastrointestinal tract, and lung explains the extracutaneous features [14,15]. However, Abu Sa’d et al. reported a case of severe lethal JEB caused by a homozygous mutation in the *COL17A1* gene [76].

#### 3.2.2. JEB, Intermediate

*JEB*, intermediate previously called JEB non-Herlitz, presents a less severe clinical phenotype than JEB severe, with a reduced tendency to develop exuberant granulation tissue. Generalized blisters (that predominate in sites exposed to friction, trauma, or heat), heal with atrophy and pigmentation anomalies. Alopecia, enamel defects, and dystrophy or absence of nails are common. Also, a milder involvement of the mucous membranes of the upper respiratory tract (with a lower risk of upper airway occlusion), bladder, and urethra was reported, and adult patients have an increased risk of developing squamous cell carcinoma on their lower extremities in areas of chronic blistering, long-standing erosions, or atrophic scarring [22,24,78]. Specific mutations in the *LAMA3*, *LAMB3*, and *LAMC2* genes (missense or splice-site or compound heterozygosity) that lead to partially functional laminin 332 are reported in this subtype. The intermediate JEB phenotype is also associated with mutations in the *COL17A1* gene. The hallmark of these phenotypes was the total lack of collagen XVII in the skin due to different mutation mechanisms (nonsense/insertions and deletions predicted to result in premature termination/splice site with the production of truncated unstable molecules). The majority of mutations were located in exons 51 and 52. Notably, splice-site mutations occurred preferentially in intron 51 [79]. In this subtype, Jonhman and Pasmooji reported the first cases with revertant mosaicism, both in collagen and laminin deficiency, sustained by the reexpression of the deficient protein on skin specimens. Revertant mosaicism has since been documented in EB forms, implicating the *COL17A1*, *KRT14*, *LAMB3*, *COL7A1*, and *FERMT1* genes [80,81]. Cases with self-improving JEB and milder than expected phenotypes were also reported. The possible underlying molecular mechanisms are an alternative modulation of splicing, a spontaneous readthrough of premature termination codons, or a skipping of exons containing stop codons [1,82,83,84].

#### 3.2.3. JEB, with Pyloric Atresia

JEB, with pyloric atresia, presents an association between generalized blistering at birth and pyloric atresia. Other gastrointestinal anomalies, such as duodenal and anal atresia, are rarely reported. Aplasia cutis congenita, atrophic scarring, enamel anomalies, oral involvement, and nail dystrophy with patulous nail folds are common. Exuberant granulation tissue in the perioral, neck, and upper back regions may occur. This disorder is associated with a significant risk of genitourinary anomalies (polypoid bladder lesions, urethral stricture, dysplastic kidney, hydronephrosis, ureterocele) and infantile or neonatal death [14,85,86]. JEB with pyloric atresia is associated with mutations in the *ITGA6* or *ITGB4* genes. The majority of the mutations reside in the *ITGB4* gene, being nonsense (with the formation of a premature stop codon) or missense mutations in the amino-terminal extracellular domain that facilitate the association of the alpha-6 or beta-4 subunits. Loss of function mutations in the *ITGA6* gene have been identified in some cases [14,53,87]. The absence of alpha-6 integrins modifies the adhesion of the collecting duct cells to the basal membrane and makes the kidney-collecting system susceptible to degeneration and injury, which explains its renourinary features [88]. DeArcangelis et al. demonstrated by alfa-6 integrin ablation in mice that loss of intestinal epithelial cells/basal membrane interactions initiates the development of inflammatory lesions that progress into high-grade dysplasia and carcinoma [89].

#### 3.2.4. JEB, Localized

Localized JEB is characterized by mild blistering, often acral, variable nail dystrophy, enamel defects, and a tendency to develop cavities. In contrast to the other JEB subtypes, alopecia, extensive atrophic scars, and extracutaneous findings are rarely reported [24,79]. Mutations in the *LAMA3*, *LAMB3*, *LAMC2*, *COL17A1*, *ITGB4*, and *ITGA3* genes were reported. Mutations allowing the expression of a residual protein (usually missense or splicing) lead to this mild phenotype. Condrat and Has stated that as little as 5–10% of residual protein, even if truncated and putatively partially functional, significantly alleviates the phenotype. Some mutations in *COL17A1* that are predicted to lead to a premature stop codon (associated with severe phenotype) escape this outcome because of alternative splicing. Out of the 56 exons of *COL17A1*, 54 are in-frame and can be skipped without shifting the reading frame [57,90,91].

#### 3.2.5. JEB, Inversa

Congenital blistering and erosions confined to flexural areas are suggestive of this rare form of JEB. Blistering is usually severe and may heal with atrophic scarring and milia formation. Nail dystrophy, enamel anomalies and dental caries, oral, esophageal, and vaginal involvement are common. Reduced expression of laminin 332 due to biallelic mutations in the *LAMA3*, *LAMB3*, and *LAMC2* genes was reported [32,92].

#### 3.2.6. JEB, Late-Onset

In contrast to the other subtype of JEB with early-onset, in JEB with late-onset (JEB-lo) the blistering starts in childhood and affects the hands and feet and, to a lesser extent, the elbows, knees, and oral mucosa. Other clinical features are palmoplantar hyperhidrosis, enamel defects, and progressive skin atrophy. The disappearance of dermatoglyphs because of scarring is reported [7]. Yuen et al. reported cases with mutations located in the fourth noncollagenous domain (NC4) of the alpha-1(XVII)-chain gene (c.3908G > A, [*p*.R1303Q or *p*.Arg1303Gln]), which are predicted to affect protein folding and laminin 332 binding. They suggested that missense mutations located in the NC4 domain may be specific for JEB-lo [78].

#### 3.2.7. Laryngo–Onycho–Cutaneous Syndrome

Laryngo–onycho–cutaneous syndrome (LOC), previously called Shabbir syndrome, is characterized by a hoarse cry in the neonatal period, by marked exuberant granulation tissue, in particular affecting the larynx, conjunctiva, and periungual/subungual sites, and by skin blistering and erosions. In contrast to the excessive blistering and erosions described in severe JEB, patients with LOC have minimal blistering but more extensive granulation tissue. Ocular granulation tissue may extend leading to symblepharon and corneal opacification (suggestive features of LOC). Progressive laryngeal granulation can lead to severe respiratory compromise and premature death. Aberrant granulation tissue could also develop on the face, neck, epiglottis, trachea, and main bronchi. Nail dystrophy and enamel anomalies are common [7,32,93,94]. In the affected members of 15 families, McLean et al. identified a homozygous single nucleotide insertion in the *LAMA3* gene (c.151dup; [V51fs]), predicting a stop codon in exon 39 that is specific to laminin alpha-3A, a protein secreted only by the basal keratinocytes of stratified epithelia. They suggested that LOC may be caused by the dysfunction of keratinocyte–mesenchymal communication and hypothesized that the laminin alpha-3A N-terminal domain may be a key regulator of the granulation tissue response. All cases reported are of Punjabi origin, suggesting a possible founder effect. Prodinger et al. reported 3 new mutations in the *LAMA3* gene, outside exon 39 and underscores that molecular diagnostics can be challenging [93,95].

#### 3.2.8. JEB, with Interstitial Lung Disease and Nephrotic Syndrome

The association of congenital nephrotic syndrome, interstitial lung disease, and skin fragility is suggestive of JEB with interstitial lung disease and nephrotic syndrome (ILNEB). The respiratory and renal features predominate and rapid progression usually leads to death in early infancy. The renal anomalies occurring in patients with ILNEB include congenital nephrotic syndrome, focal-segmental glomerulosclerosis, bilateral renal cysts, unilateral kidney hypoplasia, and ectopic conjoint kidney. Patients present variable degrees of cutaneous involvement, nail dystrophy, and sparse hair. Cases with mild phenotypes (without renal anomalies or without lung disease) were reported [34,96,97]. Biallelic mutations (missense, frameshift, or in splice sites) in the *ITGA3* gene were identified. Has et al. reported cases with functionally null mutations and a severe course of disease. Mutations allowing expression of a residual, truncated, or dysfunctional protein may lead to a milder phenotype and improved survival. Lin et al. stated that the phenotype of the *ITGA3* gene mutation may be determined by the residual function of the mutant integrin alpha-3 strain [98,99].

### 3.3. Dystrophic EB

In dystrophic EB (DEB) the plane of skin cleavage is below the lamina densa in the most superficial portion of the dermis. DEB may be inherited in a dominant (DDEB) or recessive (RDEB) pattern. The prevalence of DDEB and RDEB is quite similar: 1.49 and 1.35 per one million live births respectively [20]. In DEB, blisters, and ulcerations heal with significant scarring and milia formation. Generally, the recessive form is more severe than DDEB; however, there is significant phenotypic overlap between subtypes. All subtypes of DEB are caused by mutations in the *COL7A1* gene, the gene coding collagen VII, the main constituent of the anchoring fibrils at the cutaneous basement membrane zone [1,24,57]. Hovnanian et al. stated that the nature and location of these mutations are important determinants of the phenotype [100]. Mariath et al. suggested that the DEB phenotype is determined by the expression and residual function of collagen VII [24].

#### 3.3.1. DDEB

The majority of DDEB cases result from dominant-negative mutations. Missense substitutions that replace glycine in the collagenous triple-helical domain (frequently in exons 73, 74, and 75) are reported in over 75% of cases. The most common DDEB-causing mutations are c.6100G > A (*p*.Gly2034Arg or G2034R) and c.6127G > C (*p*.Gly2043Arg or G2043R) [101,102]. The conservation of glycine residues in every third position of the amino acid sequence is required for the tied packing of the triple helix and these substitutions highly destabilize the triple helix [103]. Other substitutions, insertions, deletions, and splice-site variants have also been described. These mutations involve amino acids essential for the structure of the triple helix and the stability of the anchoring fibrils. However, an inter- and intrafamilial phenotypic variability is reported [86,104,105].

##### Intermediate DDEB

This subtype presents with generalized blisters from birth or early infancy, milia, albopapuloid lesions, atrophic scarring, and nail dystrophy. Acral sites, elbows, and knees are commonly affected. Mucous membranes may also be involved leading to microstomia, ankyloglossia, and esophageal stenosis, although less commonly than in severe RDEB [1,24].

##### Localized DDEB

Blistering is confined to the hands, feet, and milia and atrophic scars can also occur. There is no extracutaneous involvement. Rare cases with progressive nail dystrophy and without any other sign of skin fragility are reported [106]. A pretibial form with the development of lesions predominantly in the anterior lower legs is described [107].

#### 3.3.2. RDEB

##### Severe RDEB

The most severe subtype of DEB, formerly known as Hallopeau-Siemens RDEB, is associated with generalized blistering at birth, progressive extensive scarring, and development of microstomia, ankyloglossia, esophageal stenosis, flexion contractures of limbs, and pseudosyndactyly. Alopecia, milia, and permanent loss of nail plates are common. Eye involvement with corneal erosions, symblepharon, ectropion, and loss of vision is also observed [32,108,109]. The lifetime risk of aggressive squamous cell carcinoma is greater than 90% [110]. Biallelic nonsense or frameshift *COL7A1* gene mutations (insertions/deletions, substitutions, or splice sites) that result in premature termination codons were reported. The consequences for the protein are severe: the absence of or a markedly reduced collagen VII [101,105,110,111].

##### Intermediate RDEB

Phenotype is similar to intermediate DDEB, but with greater severity of joint contractures and pseudosyndactyly in some cases. Extracutaneous involvement is milder than in severe RDEB. The risk of developing squamous cell carcinomas is also increased (47.5% by age 65) but less common than in severe RDEB and neoplasia occurs later in adulthood [7,108,110]. Many patients are compound heterozygous for a premature stop codon and a glycine substitution within the collagenous domain. The mutations may affect the association of polypeptides and the stability of the triple helix or may cause conformational change [14,86,105].

##### RDEB, Inversa

This rare subtype is characterized by a peculiar course. Generalized blistering of intermediate severity occurs in the neonatal period, improves with age, and tends to localize to flexure sites in adults. Mucosal involvement (oral, esophageal, anal, genitourinary) is similar but milder than in severe RDEB [32,112,113]. Van den Akker et al. reported specific glycine or arginine substitutions in the carboxyl portion of the triple-helical domain caused by a missense mutation in the *COL7A1* gene. Patients were homozygotes or compound heterozygotes (missense mutation/loss of function mutation). The localization of the amino acid substitutions in specific domains correlates with the synthesis of a thermolabile collagen VII that is specifically less stable in the warm flexural regions [112,113].

##### RDEB, Localized

The phenotype is similar to localized DDEB. Splice-site mutations and other amino acid (non-glycine) substitutions were reported. In localized RDEB, splice-site mutations result in exon skipping, without altering the remaining protein sequence. This abnormal collagen VII allows the assembly of the anchoring fibrils with small functional defects, which explains the phenotype [61,100,111].

#### 3.3.3. DEB, Pruriginosa

DEB, pruriginosa (*DEB-Pr*) is an unusual subtype that presents blistering in infancy and late-onset (adolescence/adulthood) of intense pruritus and linear cords of lesions (papules, nodules), especially on the extensor surfaces of the limbs (initially on the lower legs). Nail dystrophy, milia, and atrophic scarring are common [7]. Cases with autosomal dominant and autosomal recessive inheritance have been described and glycine substitutions in the collagenous domain, splice-site mutations, and small deletions have been reported. Some of these mutations have been reported in cases with other subtypes of DEB, without pruritus. No specific correlation of the genotype–phenotype has been established. Patients were shown to synthesize a normal or variably reduced amount of type VII collagen, which was correctly deposited at the dermal–epidermal junction [104,114,115,116,117]. Studies have excluded other triggering factors, including atopy, elevated IgE levels, matrix metalloproteinase 1 gene polymorphisms, filaggrin gene mutations, and interleukin 31 gene haplotypes [117,118,119,120].

#### 3.3.4. DEB, Self-Improving

Previously known as transient bullous dermolysis in a newborn, this rare subtype is characterized by generalized blistering at birth followed by significant improvement within the first 2 years of life [22]. Both dominant and recessive inheritance have been reported in cases of self-improving DEB. The most frequently reported mutations are glycine substitutions and splice-site variants resulting in the skipping of exons (e.g., exon 36). The immunofluorescence shows the accumulation of granular intraepidermal deposits of collagen VII, which regresses with time [121,122]. Christiano et al. suggested that with advancing age, the abnormal polypeptides become degraded at an increasing rate, thus diminishing their dominant-negative effects. The genotype–phenotype relationship remains unclear because of the limited number of cases [123,124,125].

#### 3.3.5. DEB, Severe, Dominant, and Recessive (Compound Heterozygosity)

The phenotype is indistinguishable from severe RDEB, with severe mucocutaneous involvement from birth. Compound heterozygosity for dominant *COL7A1* glycine substitution mutation and recessive mutation (frameshift leading to a premature termination codon) on the second allele has been reported [14,126,127,128].

### 3.4. Kindler EB

In contrast to other types of EB, Kindler EB (KEB) presents a blister formation at different levels of the dermal–epidermal junction: below the lamina densa, within the lamina lucida, or within basal keratinocytes. A single or multiple cleavage planes may be seen within the same sample of skin. KEB manifests with generalized blistering (more prominent on extremities) at birth followed by the development of photosensitivity and progressive poikiloderma. Palmoplantar keratoderma and skin atrophy may occur. Extracutaneous findings include chronic gingivitis, periodontitis, esophageal strictures, ectropion, anal stenosis, and colitis. Pseudosyndactyly has been reported. Patients with KEB have an increased risk of developing cutaneous squamous cell carcinoma (66.7% in those >60 years of age), usually occurring in the fourth to fifth decade of life [22,129]. KEB is caused by a homozygous mutation in the *FERMT1* gene. Zhang et al. suggested that fermitin family homolog 1 is also important for the suppression of UV-induced inflammation and DNA repair [130]. The protein is predominantly expressed in the epithelial cells in the skin, oral mucosa, and the gastrointestinal tract, explaining the distribution of manifestations [131,132]. Deletions, insertions, nonsense, splice-site, and missense mutations (majority loss-of-function) have been reported. Has et al. suggested that mutations compatible with the expression of an abnormal protein (e.g., in-frame) will translate into mild phenotypes, whereas null mutations cause severe forms [133].

## 4. Current Molecular Approach in Therapeutics

Molecular therapies for EB are conducted in correlation with the mutant genes and specific mutations. They are represented by gene-replacement therapies, gene editing, natural gene therapy, exon skipping, protein therapy, read-through therapies, and small molecules repurposed to relieve symptoms [134]. However, all these treatment methods are still in the phase of therapy trials. Has et al. summarize these gene therapy trials. Mainly recessive dystrophic EB and the type VII collagen and type XVII collagen proteins are targeted. There are ongoing trials in phase I/II in which interventions consist of the ex vivo grafting of gene-corrected epidermal sheets with a gamma-retroviral vector carrying *COL17A1* cDNA or *COL7A1* cDNA [134,135]. The gene-editing strategies are in the preclinical phase and use gene correction in keratinocytes or fibroblasts from patients with RDEB and the skin grafts are transplanted into immunocompromised mice. There are also studies for the JEB and *LAMB3* genes and the EBS and *KRT14* genes [134].

RNA-based therapies use antisense oligonucleotides (ASO) for in-frame exon skipping in the *COL7A1* gene. There are preclinical studies with good results in the skipping exons 13, 70, 73, 80, or 105 in the *COL7A1* gene. A clinical trial testing the ASO-targeting exon 73 in *COL7A1* is currently ongoing [134].

Protein therapy uses recombinant type VII collagen and a phase I/II clinical trial is ongoing in order to evaluate its safety and tolerability in adults with RDEB [134].

The read-through therapies use small molecular-weight compounds, which incorporate an amino acid in a place of a stop codon and in such a way as to suppress the nonsense mutations. Gentamicin was used in clinical studies for RDEB and JEB and also amlexanox to induce the read-through of *COL7A1* [134].

A small molecule used in clinical trials for the reduction of fibrosis (a major complication of RDEB) is losartan [134,136].

## 5. Conclusions

Epidermolysis bullosa is characterized by high-clinical, allelic, and locus heterogeneity. These features could be explained by the multitude of proteins that are involved in communication and signaling at the basal layers of the skin. In addition, the phenotypes are overlapping and different mutations in the same genes produce the different forms of the disease. The deciphering of pathogenic mechanisms corroborated with the discovery of the genotype–phenotype correlations and will form the basis of personalized management and the prevention of complications.

## Figures and Tables

**Figure 1 diagnostics-12-01325-f001:**
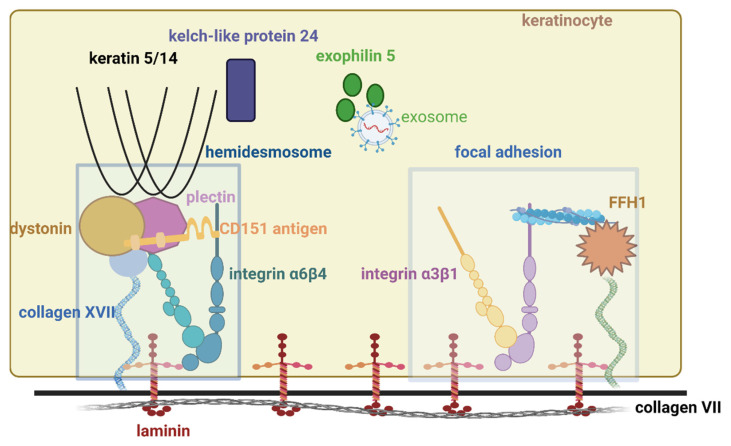
Main proteins involved in epidermolysis bullosa. Created with BioRender.com (accessed on 6 April 2022).

**Table 1 diagnostics-12-01325-t001:** Genes and proteins involved in epidermolysis bullosa [1,5,6].

Gene (Previous/ Symbol)	Approved Name (Previous/Alternative Name)	Chromosomal Location	Protein—Recommended Name (Previous/Alternative Name)	Epidermolysis Bullosa Type
***KRT5*** (*EBS2, KRT5A, CK-5*)	**keratin 5** (epidermolysis bullosa simplex 2 Dowling–Meara/Kobner/Weber–Cockayne types; keratin 5 (epidermolysis bullosa simplex, Dowling–Meara/Kobner/Weber–Cockayne types); keratin 5, type II)	12q13.13	**Keratin, type II cytoskeletal 5** (58 kDa cytokeratin; Cytokeratin-5; Keratin-5; Type-II keratin Kb5)	**EB simplex, AD** **EB simplex, AR**
***KRT14*** (*EBS3, EBS4*)	**keratin 14** (keratin 14 (epidermolysis bullosa simplex, Dowling–Meara, Koebner); keratin 14, type I))	17q21.2	**Keratin, type I cytoskeletal 14** (Cytokeratin-14; Keratin-14)	**EB simplex, AD** **EB simplex, AR**
***PLEC*** (*EBS1, PLEC1, PCN, PLTN*)	**Plectin** (plectin 1, intermediate filament binding protein, 500 kD; epidermolysis bullosa simplex 1 (Ogna); plectin 1, intermediate filament binding protein 500 kDa)	8q24.3	**Plectin** (Hemidesmosomal protein 1—Plectin-1)	**EB simplex, AD** **EB simplex, AR**
***KLHL24*** (*DRE1, FLJ20059*)	**Kelch-like family member 24** (kelch-like 24 (Drosophila))	3q27.1	**Kelch-like protein 24** (Kainate receptor-interacting protein for GluR6, Protein DRE1)	**EB simplex, AD**
***DST*** (*BPAG1, BP240, KIAA0 728, FLJ 21489, FLJ 13425, FLJ 32235, FLJ 30627, CATX-15, BPA, MACF2*)	**Dystonin** (bullous pemphigoid antigen 1, 230/240 kDa)	6p12.1	**Dystonin** (230 kDa bullous pemphigoid antigen; 230/240 kDa bullous pemphigoid antigen; Bullous pemphigoid antigen 1; Dystonia musculorum protein; Hemidesmosomal plaque protein)	**EB simplex, AR**
***EXPH5*** (*SLAC2-B*)	**exophilin 5** (synaptotagmin-like homologue lacking C2 domains)	11q22.3	**Exophilin-5** (Synaptotagmin-like protein homolog lacking C2 domains b)	**EB simplex, AR**
***CD151*** (*SFA-1, PETA-3, TSPAN24, RAPH*)	**CD151 molecule** (**Raph blood group**) (CD151 antigen; CD151 antigen (Raph blood group))	11p15.5	**CD151 antigen** (GP27; Membrane glycoprotein SFA-1; Platelet-endothelial tetraspan antigen 3; Tetraspanin-24; *CD_antigen*: CD151)	**EB simplex, AR**
***LAMA3*** (*LAMNA*; *nicein-150 kDa; kalinin-165 kDa; BM600–150 kDa epiligrin*)	**laminin subunit alpha 3** (laminin, alpha 3 (nicein (150 kD), kalinin (165 kD), BM600 (150 kD), epiligrin; laminin, alpha 3)	18q11.2	**Laminin subunit alpha-3** (Epiligrin 170 kDa subunit; Epiligrin subunit alpha; Kalinin subunit alpha; Laminin-5 subunit alpha; Laminin-6 subunit alpha; Laminin-7 subunit alpha; Nicein subunit alpha)	**Junctional EB, AR**
***LAMB3*** (*LAMNB1, nicein-125 kDa, kalinin-140 kDa, BM600–125 kDa*)	**laminin subunit beta 3** (laminin, beta 3 (nicein (125 kD), kalinin (140 kD), BM600 (125 kD)); laminin, beta 3)	1q32.2	**Laminin subunit beta-3** (Epiligrin subunit bata; Kalinin B1 chain; Kalinin subunit beta; Laminin B1k chain; Laminin-5 subunit beta; Nicein subunit beta)	**Junctional EB, AR**
***LAMC2*** (*EBR2, LAMB2T, LAMNB2, EBR2A, nicein-100 kDa, kalinin-105 kDa, BM600–100 kDa*)	**laminin subunit gamma 2** laminin, gamma 2 (nicein (100 kD), kalinin (105 kD), BM600 (100 kD), Herlitz junctional epidermolysis bullosa)); laminin, gamma 2	1q25.3	**Laminin subunit gamma-2** (Cell-scattering factor 140 kDa subunit; Epiligrin subunit gamma; Kalinin subunit gamma; Kalinin/nicein/epiligrin 100 kDa subunit; Ladsin 140 kDa subunit; Laminin B2t chain; Laminin-5 subunit gamma; Large adhesive scatter factor 140 kDa subunit; Nicein subunit gamma)	**Junctional EB, AR**
***COL17A1*** (*BPAG2, BP180*)	**collagen type XVII alpha 1 chain** (collagen, type XVII, alpha 1)	10q25.1	**Collagen alpha-1** (**XVII**) **chain** (180 kDa bullous pemphigoid antigen 2; Bullous pemphigoid antigen 2)	**Junctional EB, AR**
***ITGA6*** (*CD49f*)	**integrin subunit alpha 6** (integrin, alpha 6)	2q31.1	**Integrin alpha-6** (CD49 antigen-like family member F VLA-6; *CD_antigen*: CD49f)	**Junctional EB, AR**
***ITGB4*** (*CD104*)	**integrin subunit beta 4** (integrin, beta 4)	17q25.1	**Integrin beta-4** (GP150, *CD_antigen*: CD104)	**Junctional EB, AR**
***ITGA3*** (*MSK18, CD49c, VLA3a, VCA-2, GAP-B3*)	**integrin subunit alpha 3** (antigen identified by monoclonal antibody J143; integrin, alpha 3 (antigen CD49C, alpha 3 subunit of VLA-3 receptor))	17q21.33	**Integrin alpha-3** (CD49 antigen-like family member C, FRP-2; Galactoprotein B3; VLA-3 subunit alpha; *CD_antigen*: CD49c; CD49 antigen-like family member C)	**Junctional EB, AR**
***COL7A1*** (*EBDCT, EBD1, EBR1*)	**Collagen type VII alpha 1 chain** (epidermolysis bullosa, dystrophic, dominant and recessive; collagen, type VII, alpha 1; collagen VII, alpha-1 polypeptide; LC collagen)	3p21.31	**Collagen alpha-1** (**VII**) **chain** (Long-chain collagen)	**Dystrophic EB, AD** **Dystrophic EB, AR**
***FERMT1*** (*C20orf42, FLJ20116, URP1, KIND1, UNC112A*)	**FERM domain containing kindlin 1** (chromosome 20 open reading frame 42; fermitin family homolog 1 (Drosophila); fermitin family member 1; kindlin-1; kinderlin)	20p12.3	**Fermitin family homolog 1** (Kindlerin; Kindlin syndrome protein, Kindlin-1; Unc-112-related protein 1)	**Kindler EB, AR**

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
