# Peer review of "Epidermolysis Bullosa—A Different Genetic Approach in Correlation with Genetic Heterogeneity"

_diagnostics, 2022, doi:10.3390/diagnostics12061325_

Round 1

Reviewer 1 Report

This is a detailed and exhaustive review of a extremely rare disease, but that every physician should be aware of. Gene mutations and their implications in clinical manifestation have been widely analyzed. Although I recognize that the treatment is not the main topic of this review, I would add some short information about the phase I and II trials of genetically corrected autologous epidermal grafts.

Author Response

“This is a detailed and exhaustive review of a extremely rare disease, but that every physician should be aware of. Gene mutations and their implications in clinical manifestation have been widely analyzed. Although I recognize that the treatment is not the main topic of this review, I would add some short information about the phase I and II trials of genetically corrected autologous epidermal grafts.”

Answer: Thank you very much for your comments and constructive observations.

We added a small section on current therapies (“Current molecular approach in therapeutics”)

Reviewer 2 Report

A narrative review about the genetic characteristics of epidermolysis bullosa; The paper may require further revisions in order to be considered for publication: clinical characteristics of EB should be better assessed in the introduction, as well as histopathological findings.

A small section on current therapies would also be a good addition to the paper.

Thank You

Author Response

“A narrative review about the genetic characteristics of epidermolysis bullosa; The paper may require further revisions in order to be considered for publication: clinical characteristics of EB should be better assessed in the introduction, as well as histopathological findings.

A small section on current therapies would also be a good addition to the paper.

Thank You”

Answer: Thank you very much for your comments and constructive observations.

We added clinical characteristics of EB and histopathological findings in the introduction section.

We added a small section on current therapies (“Current molecular approach in therapeutics”).

Reviewer 3 Report

This review describes the genetic heterogeneity in Epidermolysis Bullosa (EB). The manuscript is clearly written and summarizes to the current knowledge about EB with respect to mutated genes.

Minor issues:

The reference to table I is written bold (line 44/45). I guess this is a mistake.

Throughout the manuscript, the authors abbreviate and others using 'et al'. This is not correct and should be change to 'et al.'.

Author Response

“This review describes the genetic heterogeneity in Epidermolysis Bullosa (EB). The manuscript is clearly written and summarizes to the current knowledge about EB with respect to mutated genes.

Minor issues:

The reference to table I is written bold (line 44/45). I guess this is a mistake.

Throughout the manuscript, the authors abbreviate and others using 'et al'. This is not correct and should be change to 'et al.'.”

Answer: Thank you very much for your comments and constructive observations.

We modified line 44/45 without bold. We replaced in the manuscript “et al” with “et al.”